# Enhancing Health and Empowerment: Assessing the Satisfaction of Underprivileged Rural Women Participating in a Functional Literacy Education Program in Kailali District, Nepal

**DOI:** 10.3390/healthcare12111099

**Published:** 2024-05-27

**Authors:** Joong Seon Na, Johny Bajgai, Subham Sharma, Sarmila Dhakal, Dong Won Ahn, Young-Ah Doh, Yundeok Kim, Kyu-Jae Lee

**Affiliations:** 1Department of Convergence Medicine, Wonju College of Medicine, Yonsei University, Wonju 26426, Gangwon-do, Republic of Korea; estherna@yonsei.ac.kr (J.S.N.); johnybajgai@yonsei.ac.kr (J.B.); 2020323346@yonsei.ac.kr (S.S.); 2Department of Public Health, National Academy of Medical Science, Purbanchal University, Biratnagar 56600, Nepal; sarmiladhakal75@gmail.com; 3Institute for Poverty Alleviation and International Development, Yonsei University, Mirae Campus, Wonju 26493, Gangwon-do, Republic of Korea; dongwona@yonsei.ac.kr; 4Evaluation Department, Korea International Cooperation Agency, 825 Daewangpangyo-ro, Sujeong-gu, Seongnam-si 13449, Gyeonggi-do, Republic of Korea; yadoh@koica.go.kr; 5Department of Internal Medicine, Division of Hematology-Oncology, Wonju Severance Christian Hospital, Wonju 26426, Gangwon-do, Republic of Korea; 6Department of Global Medical Science, Wonju College of Medicine, Yonsei University, Wonju 26426, Gangwon-do, Republic of Korea

**Keywords:** illiteracy, functional literacy education program, health literacy, rural women, underprivileged

## Abstract

Women’s empowerment and health literacy are essential for fostering community well-being. Empowering women through education and diverse training plays a crucial role in ensuring their prosperity and overall health. This study investigates the satisfaction and experiences of underprivileged rural mothers participating in a functional literacy education program in the Kailali district, Nepal. We assess participants’ perceptions of program effectiveness, examining training content, facilities, and trainers while exploring menstrual hygiene practices and maternal health awareness. Through convenience sampling, 141 underprivileged women from five rural villages near Tikapur were selected from literacy centers run by Mahima Group. Utilizing structured questionnaires and statistical analyses, including descriptive analyses, Spearman’s rho correlation, and Pearson’s chi-square test, we found that 65.2% of participants expressed high satisfaction levels. Moreover, 96.5% found the program highly effective, with 97.9% reporting improved literacy skills and 96.5% demonstrating increased awareness of menstrual hygiene practices. Additionally, 97.2% agreed that the program enhanced maternal and child health knowledge. Significant correlations were observed among the training course, facilities, trainers, and overall training perception. In line with this, significant associations were found between age groups (*p* = 0.003) and geographical areas (*p* = 0.023) with satisfaction levels with the literacy program. These results underscore the satisfaction of participants within the literacy program and its impact on their lives, and advocates for its broader implementation to empower marginalized communities for sustainable development.

## 1. Introduction

Education stands as a potent tool for fostering societal transformation and is universally acknowledged as an essential human right applicable across a person’s lifespan. It is widely recognized as a cornerstone for achieving sustainable development and holds profound significance in the context of women’s empowerment [1]. Furthermore, education plays an integral role in fostering not only physical well-being but also mental faculties such as critical reasoning, emotional regulation, and adept social interaction skills [2]. Additionally, education has been identified as a pivotal determinant for fostering adequate health literacy and, consequently, promoting overall well-being. Health literacy is defined as “personal knowledge and competencies that empower individuals to access, comprehend, evaluate, and utilize information and services to foster and sustain good health and well-being for themselves and their communities” [3]. Studies have reported that globally, more than 80% of individuals living in poverty reside in rural areas, with approximately 70% of this demographic being women. Furthermore, on a global scale, the number of illiterate individuals exceeds 750 million, with nearly half of this population concentrated in South Asian countries [4,5]. Women’s education plays a pivotal role in developing a sense of self-worth, improving knowledge, creating good practices for public health awareness, and strengthening family bonds and social relationships, which are important for a country’s economic and sustainable development [6,7]. However, many women in several low- and middle-income countries remain deprived of a formal education [7]. To alleviate these problems, such people must have alternative opportunities to receive education. A previous study reports that knowledge can be achieved through three different modes: formal, non-formal, and informal [8]. In numerous cases, the non-formal education approach is recognized as a complementary framework to formal education, with functional literacy education programs serving as a prominent example. These programs are specifically structured to aid adults in navigating the practical challenges encountered in their daily lives. Studies have demonstrated functional adult literacy programs as holistic initiatives designed to provide individuals with vital skills such as reading, writing, numerical calculation, and basic vocational competencies. These programs address various aspects, including family literacy, health promotion, empowerment, and economic self-sufficiency. Additionally, by targeting both literacy and livelihood goals, these types of programs play a crucial role in improving economic, societal, and family-related outcomes, particularly for vulnerable populations. Moreover, these programs serve as vital strategies for empowering individuals living in poverty, facilitating livelihood enhancement and social mobility [9,10,11,12]. In addition, functional adult literacy programs are recognized for their pivotal role in enhancing cognitive development, fostering critical thinking and problem-solving skills essential for adopting healthy behaviors [13]. They contribute to shaping health literacy directly through participation in health literacy courses and indirectly through the promotion of literacy, numeracy, communication, and social skills. Proficiency in reading, writing, advanced cognitive abilities, and social skills are essential prerequisites for achieving health literacy. Research indicates that functional adult education positively influences the understanding of health information and related systems by improving communication and writing skills. Furthermore, it fosters increased self-esteem and self-efficacy, crucial for effectively coping with challenging situations and facilitating successful health behavior change [14,15,16,17,18]. A previously published study indicated that adults lacking literacy skills often experience poorer health outcomes and have limited engagement with their communities [19]. Conversely, another study highlighted that participants who participated in functional education programs exhibited increased vocalization in effecting societal change and demonstrated a greater capacity to enhance their socioeconomic circumstances [20].

Unlike many low- and middle-income countries, Nepali women have low-level access to education, healthcare, and socioeconomic opportunities, especially in the rural parts of the country [21]. Within Nepal’s patriarchal societal structure, women often lack autonomy, contend with discrimination, and contend with male-dominated hierarchies. Traditionally, women bear the dual responsibilities of caregiving for their families and managing household tasks [22]. Consequently, they are frequently marginalized in terms of educational opportunities. Of these, it has been reported that the male literacy rate in the country is significantly higher (76.4%) than that of females (53.1%). These data show a huge disparity in literacy rates between both genders in Nepal [23,24]. Additionally, studies have suggested that menstrual hygiene practices are poor in many rural areas of the Far-Western Development Region of Nepal, also known as Sudurpaschim province. Menstrual exile in this region is popularly known as chhaupadi (menstruation women), in which poor menstrual hygiene is practiced, as menstruating women and girls traditionally reside in unhygienic sheds for four consecutive days, which is unsafe and lacks basic necessities. Consequently, women suffer from several health issues, including urinary and reproductive tract infections [25,26]. This social taboo is exacerbated in these societies due to prevalent illiteracy and lack of access to health-related knowledge. In Nepal, maternal and neonatal health remains a significant public health concern, with an estimated maternal mortality rate of 239 per 100,000 live births and a neonatal mortality rate of 21 per 1000 live births [27]. However, the utilization of maternal and neonatal health services faces various barriers, including inadequate education on maternal and child health, low health literacy, cultural norms, financial constraints, and societal disparities [28,29]. Particularly, underprivileged rural women encounter numerous challenges accessing maternal healthcare services due to factors such as caste identity and economic deprivation. For instance, women from Dalit caste groups in Nepal exhibit lower rates of maternal health service utilization [30,31]. To address these barriers, health promotion and education are essential strategies for improving maternal health outcomes, along with advancing the status of women within society [32,33]. One study in Nepal reported that mothers’ educational levels determined and influenced the utilization of maternal health services in rural communities [34]. Educating women and facilitating their mobility can play a pivotal role in fostering innovation, while also nurturing confidence in engaging with broader segments of society [32]. Therefore, in developing nations like Nepal, it is imperative to tailor functional literacy education programs to address the specific contextual needs of learners. This approach ensures that programs are not only more effective but also more relevant and impactful on the lives of participants. Previous studies have indicated that functional literacy programs enhance adult learners’ innovative and creative skills by reshaping their contexts and life objectives through engaging in productive activities [35,36].

Nepal’s Sudurpaschim province (Province 7) is notably the most underdeveloped area among all five development regions in Nepal, as highlighted by the Human Development and Poverty Index [37]. Access to healthcare and education is particularly deficient here, with poverty representing a pervasive challenge [38,39]. Moreover, additional factors such as chhaupadi practices and limited access to maternal and child health facilities further exacerbate the hardships faced by residents [25,26,40]. Additionally, women in this region confront persistent challenges including illiteracy, domestic violence, and restricted rights [41]. Approximately, half of the female population in this region lack basic literacy skills, and less than 5 percent own property, including land or homes. A significant portion of the population belongs to marginalized and indigenous communities, including the Tharu and Dalit ethnic groups [42,43]. In light of these profound challenges, our collaboration with the Mahima Group, a local non-governmental organization (NGO) located in Tikapur, Kailali, Nepal, assumes critical importance. We shared a dedication to addressing pressing issues in the Sudurpaschim province faced by marginalized women—especially mothers—in five distinct rural villages within the periphery of the Tikapur region situated in the Kailali district of this province. The Mahima group is dedicated to empowering women from marginalized and underprivileged communities, including the Tharu and Dalit ethnic groups, through functional literacy education, awareness programs, and income-generating projects, facilitated by social mobilization and collective participation. Through collaborative efforts and community participation, we aspire to contribute to the advancement of these regions in terms of socio-development and health awareness.

Therefore, this study aimed to investigate participants’ satisfaction and experiences regarding their participation in a functional adult literacy education program and its outcomes on enhancing their livelihoods. Additionally, the researchers aimed to verify correlations between measures of the training program. To achieve this objective, the researchers intended to answer the following basic research questions:(i)What is the level of satisfaction among underprivileged rural women participating in functional literacy programs?(ii)How effective are literacy programs in improving literacy skills and the perceptions of participants regarding the effectiveness of the functional literacy education program in terms of training courses, facilities, and trainers?(iii)What impact do literacy programs have on health knowledge, particularly menstrual hygiene and maternal health, and practices of underprivileged rural women in the Kailali district, Nepal?

## 2. Materials and Methods

### 2.1. Study Area, Study Design, and Selection of Participants

This cross-sectional study was conducted in five neighboring villages (Nuklipur, Bankatti, Chamelipur, Durgauli, and Laikpur) within a 5 km radius of the Tikapur region, located in the Kailali district of the Sudurpaschim province (Figure 1), Nepal. According to a report by the Central Bureau of Statistics (2021), the Kailali district has a population of 904,666 (men: 433,456; women: 471,210), with approximately 51.97% residing in rural areas. Among the female population, approximately 40.05% were reported as illiterate [44]. The region is predominantly marked by poverty and inhabited by marginalized communities [42,43].

The Mahima Group, a local non-profit NGO operating in this area, focuses on the welfare of underprivileged groups such as the Dalit, Tharu, ex-Kamaiya families, women—particularly mothers—and the impoverished. The functional literacy education program was initiated by the Mahima Group in collaboration with our team as a pilot program from December 2013 to December 2015 and was implemented in five neighboring villages within a 5 km radius of the Tikapur region—Nuklipur, Bankatti, Chamelipur, Durgauli, and Laikpur. These villages are home to many women from underprivileged communities, characterized by poverty, illiteracy, and limited awareness of menstrual hygiene and maternal health. Recognizing this need, the Mahima Group established functional literacy centers in these areas with the goal of educating underprivileged women and promoting health awareness among them. Hence, these villages were selected as the focus areas for our study.

A total of 255 underprivileged illiterate women participated in this program across five different centers from December 2013 to December 2015. The literacy course was divided into two parts: a six-month basic literacy course focusing on fundamental reading, writing, and numeracy skills, along with confidence-building exercises that provided women with a platform to address daily challenges and advocate against injustice. This was followed by a six-month post-literacy course, which reinforced the basics of reading, writing, and counting skills while incorporating practical exercises and discussions on day-to-day issues in their work, family, and community, with an emphasis on problem-solving approaches. Classes were held six days a week, customized to the participants’ convenience, and facilitated discussions between participants and trainers. Each class session was divided into four segments: the first for reviewing and reflecting on the previous day’s learning, the second for discussing new topics relevant to their experiences and challenges, the third for introducing and practicing new vocabulary, and the fourth for developing action plans and strategies for awareness campaigns and community outreach programs. The reading and learning materials utilized were contextualized to local settings, drawing upon real-life experiences, successes, and challenges.

In our current study, 141 out of the 255 participants who engaged in both the basic and post-literacy courses of the literacy education program voluntarily participated. We employed a convenience sampling technique [45,46], selecting individuals who were readily available and willing to take part. The inclusion criteria for participants were as follows: (i) participants must have completed both the basic and post-literacy courses of the functional literacy education program, and (ii) they had to be residing in any of the five villages included in this study, being available during the survey period and willing to participate voluntarily. Data collection was supervised by a well-trained researcher, focusing on participants above the age of 18. Prior to data collection, all participants were briefed on the survey’s purpose and informed written consent was obtained from participants.

### 2.2. Data Collection Instruments and Measurements

In this study, the researchers developed a comprehensive questionnaire and conducted a survey in February 2024 to gather data from the participants. The questionnaire was structured around a five-point Likert scale and was organized into four main sections. (I) Training courses; this part focused on evaluating participants’ opinions regarding the effectiveness, relevance, and comprehensiveness of the training courses offered as part of the literacy program. (II) Facilities; this part focused on obtaining participants’ feedback based on the adequacy and accessibility of the facilities provided during the literacy program, including infrastructure, materials, and resources. (III) Trainers; this part aimed to assess participants’ perceptions of the trainers’ competency, communication skills, and ability to effectively deliver the content of the literacy program. (IV) General overview of the literacy program; in this part, participants were asked to give an overall assessment of the literacy program, including its outcomes on their lives, challenges faced, and suggestions for improvement.

In addition to the Likert scale-based questions, a structured questionnaire approach was adopted. This encompassed a range of topics to gather comprehensive data from the participants, such as their motivation and purposes for participating in the literacy program; their satisfaction levels with various aspects of the program; and the impact on the participants’ knowledge and practices related to menstrual hygiene, maternal health, and child care, in which they were encouraged to share their views on potential improvements to the literacy education program, offering valuable feedback for future enhancements. The survey questionnaire was drafted in English, translated into the Nepali language by a native speaker, and the translation quality was assured by two native speakers. Participants were individually presented with the questionnaire by a trained researcher. Each question was explained thoroughly to ensure clarity, and participants were given ample time to respond. The researcher facilitated the data collection process, ensuring accuracy and consistency in responses. Participants were encouraged to provide honest and detailed feedback. The primary objective of the survey was to gather quantitative information from participants to evaluate the outcomes of the functional adult literacy education program on their daily lives. By collecting data on various aspects of the program, including participant satisfaction, perceived effectiveness, and areas for improvement, this study aimed to provide insights into the program’s efficacy and inform future interventions.

### 2.3. Research Ethics

Prior to data collection, ethical considerations were carefully addressed. Informed consent was obtained from all participants, ensuring voluntary participation and confidentiality of responses. Participants were assured that their participation was anonymous and would not have any adverse consequences. This study was conducted in accordance with the Declaration of Helsinki and with the approval of the Ethics Committee of Nepal Health Research Council (Reg.no. 633/2023).

### 2.4. Data Analysis

After the data collection process, the researchers checked the completed questionnaires for completeness and precision. To test the reliability of the question items, the researchers calculated Cronbach’s α as the reliability index for participants’ satisfaction survey with a total of four parameters (15 questions). The responses to the structured questionnaire were quantitatively analyzed using the Statistical Package for Social Sciences (SPSS) version 25. Additionally, satisfaction with the functional literacy education program parameters was analyzed through Spearman’s rho correlation. The association between sociodemographic variables and satisfaction with functional literacy program by participants was analyzed through the Pearson chi-square test. Differences were considered to be statistically significant at *p <* 0.05. All other results are summarized using frequencies and percentages, as presented in the tables.

## 3. Results

### 3.1. Baseline Characteristics of Participants

Data were collected from 141 participants from five different villages near the Tikapur region: Nuklipur, Bankatti, Chamelipur, Durgauli, and Laikpur in the Sudurpaschim province, Nepal as shown in Table 1.

The mean age of participants was 51.18 years, with a standard deviation of 10.15 years, indicating a moderate age range within the sample. The majority of participants identified as belonging to the Tharu caste (73%), followed by the Nepali caste (27%). Regarding religious affiliation, Hinduism was predominant among participants, with 91.5% identifying as Hindu and 8.5% as Christian. Geographically, participants were distributed across different areas, with Durgauli having the highest representation (35.5%), followed by Nuklipur (33.3%), Chamelipur (20.6%), Laikpur (6.4%), and Bankatti (4.3%) Table 1.

### 3.2. Women Participants’ Response Regarding the Purpose of Participation in the Functional Literacy Education Program

Table 2 displays the varied motivations of participants participating in the literacy program. The majority (72.3%) pursued personal growth, while 7.1% aspired to have happier home lives. Additionally, 19.1% aimed to contribute to community development, with a smaller proportion (1.4%) prioritizing work–life improvement. These findings underscore the diverse reasons individuals engage in the program, emphasizing personal and community development.

### 3.3. Women Participants’ Opinions toward Satisfaction with the Functional Literacy Education Program

To validate the reliability of the questions related to satisfaction with this education program, the researchers calculated Cronbach’s (0.821) α based on standardized items as the reliability index (Appendix A) which would validate the reliability of the question items presented at Table 3.

Participants’ satisfaction with the functional literacy education program was assessed through a survey. The majority rated the training course objectives and contents positively, with 46.1% indicating a “High” usefulness and 29.1% rating it as “Very High”. Regarding income impact, 50.4% perceived a “High” impact. Participants also found the facilities and trainers satisfactory, with high ratings for convenience (50.4% “High”, 36.9% “Very High”) and trainer competence (65.2% “High”, 22% “Very High”). Overall, participants expressed a strong need for more similar training courses (70.2% “High”) and reported high satisfaction levels (44.7% “High”) with the program’s helpfulness in their lives (Table 3). These results highlight positive perceptions and demand for the training program.

### 3.4. Women Participants’ Responses to the Functional Literacy Education Program in Empowering Livelihood

Table 4 illustrates survey results showing a strong accord among participants regarding the effectiveness of the education program (96.5%). The findings also indicate high levels of future participation intent (69.5%) and program recommendation (88.7%) by participants. Moreover, participants reported positive life changes (91.5%), with 52.5% initiating income-related work and community development impact (99.3%), highlighting the program’s success in enhancing literacy skills and empowering individuals.

### 3.5. Women Participants’ Responses to the Functional Literacy Education Program Regarding Knowledge about Menstrual Hygiene

The results in Table 5 demonstrate participants’ understanding and practices related to menstrual hygiene. Furthermore, the results indicate a high level of knowledge about menstrual hygiene (96.5%) among participants, and many did not practice chhaupadi (85.8%). Additionally, a significant proportion agreed that hygiene neglect during menstruation could lead to illness (68.8%) and that clean water and soap are essential for menstrual hygiene (73%). Participants also emphasized the importance of handwashing before genital cleaning to prevent reproductive infection (71.6%) and the necessity of clean sanitary pads/reusable cloths for menstrual hygiene (57.4%). Furthermore, the majority used soap and water to clean reusable cloths (78.7%) and disposed of menstrual materials in bins (75.2%). Importantly, 94.3% of participants reported that the literacy program improved their knowledge of menstrual hygiene, highlighting its effectiveness in promoting health education and awareness.

### 3.6. Women Participants’ Responses to the Functional Literacy Education Program Regarding Knowledge about Maternal and Child Health

Table 6 demonstrates and sheds light on participants’ knowledge and practices concerning maternal and child health. A majority of participants were aware of the frequency of minimum antenatal care (ANC) visits (81.6%) and pregnancy risk symptoms (84.4%). Moreover, a high proportion of participants knew about the importance of taking supplement medicine during pregnancy (92.2%) and consuming balanced diets during and after pregnancy (96.5%). Additionally, most participants were aware of the suitable age for pregnancy (90.8%) and the importance of child immunization (76.6%). Furthermore, a significant number of participants acknowledged that the literacy program helped improve their knowledge of maternal and child health (97.2%), emphasizing its effectiveness in promoting health education. Lastly, a majority recognized the importance of handwashing before eating and after using toilets (100%). These findings underscore the program’s positive impact on participants’ awareness and practices related to maternal and child health, highlighting its role in promoting health and hygiene education.

### 3.7. Correlations (Spearman’s Rho) between the Measures of Satisfaction Level of Functional Adult Literacy Education Program Based on Training Course, Facilities, Trainers, and General Overview of Training

The correlation indicates Spearman’s rank correlation coefficients between four parameter variables: training course (TC), facilities (F), trainers (T), and general overview of training (GOT). The table shows the strength and significance of the correlations (Table 7). All correlations are statistically significant at the 0.01 level, indicating that there are significant relationships between the pairs of variables. Our results suggest that as one variable increases, the other tends to increase as well, and vice versa. The strength of the correlations is considered moderate based on the magnitude of the coefficients. These findings suggest that enhancing one aspect of the training program may positively influence other related components, contributing to overall training effectiveness and participant satisfaction.

### 3.8. Association between Sociodemographic Variables and Satisfaction with Functional Literacy Program by Participants

Table 8 illustrates the association between demographic variables and participants’ satisfaction with the literacy program. Significant associations were found between age groups (*p* = 0.003) and geographical areas (*p* = 0.023) with satisfaction levels. Participants aged 41–60 showed the highest satisfaction (60.9%), and those from Durgauli reported the highest satisfaction (58%). These findings suggest that age and geographical location may influence participants’ perceptions of the program. However, no significant associations were observed based on caste or religion.

## 4. Discussion

The current study explores the satisfaction levels and experiences of underprivileged mothers residing in rural villages in the Kailali district, while also assessing the impact of a functional education literacy program on their lives. Furthermore, it examines participants’ perceptions of the program’s effectiveness, including aspects such as training course content, facilities, trainers, and overall training experience. Additionally, the study explores participants’ menstrual hygiene practices, knowledge, awareness, and understanding, as well as maternal health-related factors influenced by the program.

The empowerment of rural women is one of the most important initiatives in recent days. Education empowers and assists women in gaining self-worth, overcoming their shortcomings, and enables them to challenge entrenched harmful traditions that contradict principles of fairness and justice [47]. A previous study conducted by Robinson-Pant emphasized that adult literacy programs not only enhance individuals’ practical skills but also empower them to lead more fulfilling lives [48]. Similarly, Rehman concluded that engagement in adult literacy programs not only cultivates emotional and psychological control in learners but also nurtures compassionate attitudes, enhances day-to-day learning, and fosters community participation [49]. The results of the study revealed that participants who participated in the functional education literacy program showed high satisfaction response, citing its suitability and usefulness in meeting their needs. This indicates that the objectives and content of the program resonated well with the participants, addressing their specific challenges and aspirations. Moreover, participants reported tangible benefits from the program, particularly in terms of improving their income-generating activities. This suggests that the literacy skills obtained from the program enabled them to explore new opportunities or enhance existing livelihood strategies, contributing to their economic empowerment. Furthermore, participants expressed satisfaction with the learning facilities and environments provided. Access to conducive learning spaces and resources is crucial for effective learning outcomes, and the positive satisfaction response in this regard reflects the program’s commitment to ensuring a supportive educational environment. Additionally, participants found the lecture contents of the education program to be both interesting and easily understandable. This indicates that the program effectively delivered educational content in a manner that engaged participants and facilitated comprehension, thereby maximizing learning outcomes. Beyond skill acquisition, participants highlighted the program’s role in fostering critical thinking and creativity. This aspect is significant as it indicates that the program not only imparted knowledge but also empowered participants to apply their learning in innovative ways to address challenges and pursue opportunities in their daily lives. This positive feedback from participants underscores the effectiveness of the functional literacy program in empowering underprivileged rural mothers. By addressing their educational needs, enhancing livelihood opportunities, and nurturing critical thinking and creativity, the program has made significant strides in improving the lives of its participants and fostering sustainable development in rural communities. Several studies have shown that participatory learning models in functional literacy education programs effectively improve learning motivation, attitudes, and achievements [13,50]. Additionally, research indicates that participatory-based functional literacy education programs positively impact learning effectiveness, measured through the achievement of objectives, participants’ engagement, knowledge acquisition, attitude improvements, and skill enhancement [1,13]. In addition, the participants expressed satisfaction with the positive outcomes of the program on their livelihoods and personal empowerment. Many reported acquiring basic literacy skills, such as reading and writing, which significantly boosted their confidence. They noted that the program facilitated positive changes in various aspects of their lives and contributed to the improvement and development of their communities. Moreover, they expressed willingness to participate in similar programs in the future and were keen to recommend them to their neighbors. These findings align with the existing literature, underscoring the transformative role of functional literacy education in empowering women and enabling them to play more active roles in social welfare activities for societal advancement [9].

In many parts of Nepal, menstrual behaviors are influenced by socio-cultural restrictions and taboos, hindering menstrual hygiene management (MHM) [51,52]. This inhibition, stemming from cultural taboos and misconceptions, particularly affects women’s self-respect, health, and education [53]. Studies have highlighted the risk of infections due to inadequate menstrual hygiene practices, leading to various health issues such as reproductive tract infections, genitourinary tract infections, and cervical cancer. Furthermore, societal taboos and misconceptions surrounding menstruation perpetuate gender inequality and hinder women’s empowerment, compounded by a lack of information and guidance on menstruation management from teachers [26,51]. In many parts of the Sudurpaschim province, including our study area, the practice of chhaupadi persists [26]. Research indicates a significant association between menstrual hygiene practices and mothers’ educational status, with mothers who are literate positively influencing their daughters’ MHM practices through prior orientation [54,55]. In our study, participants emphasized crucial hygiene practices, such as handwashing and proper disposal methods, indicating the literacy program’s success in enhancing health education and awareness. Participating in such programs likely has several positive social impacts. The increased knowledge and awareness of menstrual hygiene practices among participants suggest potential improvements in their overall health and well-being. Additionally, the reported decrease in the practice of chhaupadi in our study, although not directly assessed for causality, indicates a positive shift away from harmful cultural practices. Additionally, participants demonstrated excellent knowledge of hygiene practices, menstrual hygiene management, and waste management. Therefore, our findings emphasize the significance of functional education programs in the Sudurpaschim province for enhancing menstrual hygiene practices, illustrating their potential to empower women by fostering knowledge and positive behaviors in their day to day life and for societal changes. However, further in-depth education initiatives and broader societal interventions are necessary to tackle such complex issues effectively.

Furthermore, the program’s focus on maternal and child health education which may have contributed to increased knowledge and awareness among participants, potentially leading to better health outcomes for themselves and their families. Specifically, participants showed increased knowledge regarding the frequency of antenatal care checkups, identification of pregnancy risk symptoms, and the importance of prenatal supplement intake, such as iron, zinc, and folic acid. Additionally, they demonstrated improved awareness of maintaining a balanced diet during and after pregnancy, understanding the appropriate age for marriage, the significance of child immunization, and the importance of hygiene practices. Overall, these findings underscore the program’s potential to foster positive social change by empowering participants with knowledge and skills that can enhance their lives and contribute to the well-being of their communities. Of these, one of the studies reported that participatory women’s literacy groups were found to be effective in improving maternal and child health outcomes, along with other benefits such as increased knowledge and empowerment of family members [56]. Education programs like this can serve as effective tools in improving maternal and child health knowledge among underprivileged rural women, potentially contributing to reducing maternal and neonatal mortality rates in Nepal.

While our study offers valuable insights into the impact of functional literacy education on the lives of underprivileged women in the Sudurpaschim province, it is important to acknowledge several limitations. Firstly, the nine-year gap between the literacy program’s implementation and the administration of the questionnaire to assess the satisfaction of participants participating in a functional literacy education program poses a limitation on the accuracy and relevance of their responses. Despite this, our study underscores the transformative power of literacy programs in enhancing education, health knowledge, and empowerment among underprivileged rural women in Nepal. The remarkably positive feedback from participants demonstrates the lasting impact of these programs, providing crucial insights for policymakers, NGOs, and community organizations. This highlights the need for ongoing and timely evaluations to maximize the benefits of such initiatives in the community. Addressing this gap in future research can build on our findings to advocate for the broader implementation of literacy programs, ultimately contributing to sustainable development and the empowerment of marginalized communities in Nepal, especially in rural settings. However, we recognize the necessity for further education initiatives and broader societal interventions to address complex issues like chhaupadi. Moreover, our study serves as a step towards empowering women and promoting functional literacy education that integrates health topics such as menstrual and maternal health in Nepal. Additionally, the small sample size and convenience sampling technique used in our study may limit the generalizability of our findings and introduce potential bias, as participants were selected based on availability from five small villages near the Tikapur region. Our reliance on self-reported data may introduce social desirability bias, potentially leading to inaccuracies in the data. The absence of a control group limits our ability to compare outcomes and ascertain the specific effects of the functional literacy program. Lastly, the cross-sectional nature of our study design restricts our ability to establish causal relationships between variables. Future research should employ larger, randomized samples, incorporate objective measures to validate self-reported data, include control groups to strengthen causal inferences, and utilize longitudinal designs to explore the long-term impacts of literacy interventions on women’s empowerment and well-being.

## 5. Conclusions

In conclusion, our study underscores the significant impact of functional literacy programs on the lives of underprivileged women in rural areas, particularly in the Kailali district. Through participation in these programs, women exhibited enhanced knowledge and awareness across various domains, including general literacy skills, menstrual hygiene practices, and maternal and child health. The findings indicate a positive connection between program participation and improvements in participants’ understanding of key health topics, such as antenatal care, pregnancy risk symptoms, and nutrition during pregnancy. Moreover, the program’s emphasis on literacy and life skills has empowered women to make informed decisions and pursue opportunities for personal and economic development. Furthermore, our study highlights the importance of educational interventions in addressing the unique needs of marginalized communities in remote settings, offering a pathway to break the cycle of poverty and promote social inclusion and sustainable development. We believe that our findings serve as a valuable contribution to empowering women and promoting menstrual and maternal health in Nepal, contributing to a growing database of knowledge in this field.

However, there is a need for continued investment in functional literacy programs and similar community-based interventions to support the empowerment of women and marginalized groups. By addressing educational disparities and promoting health literacy, these initiatives can contribute to the achievement of broader development goals and create lasting positive change in rural communities. Ultimately, our study reaffirms the importance of education as a catalyst for empowerment and social transformation, emphasizing the critical role of functional literacy programs in promoting equity, dignity, and well-being for all.

## Figures and Tables

**Figure 1 healthcare-12-01099-f001:**
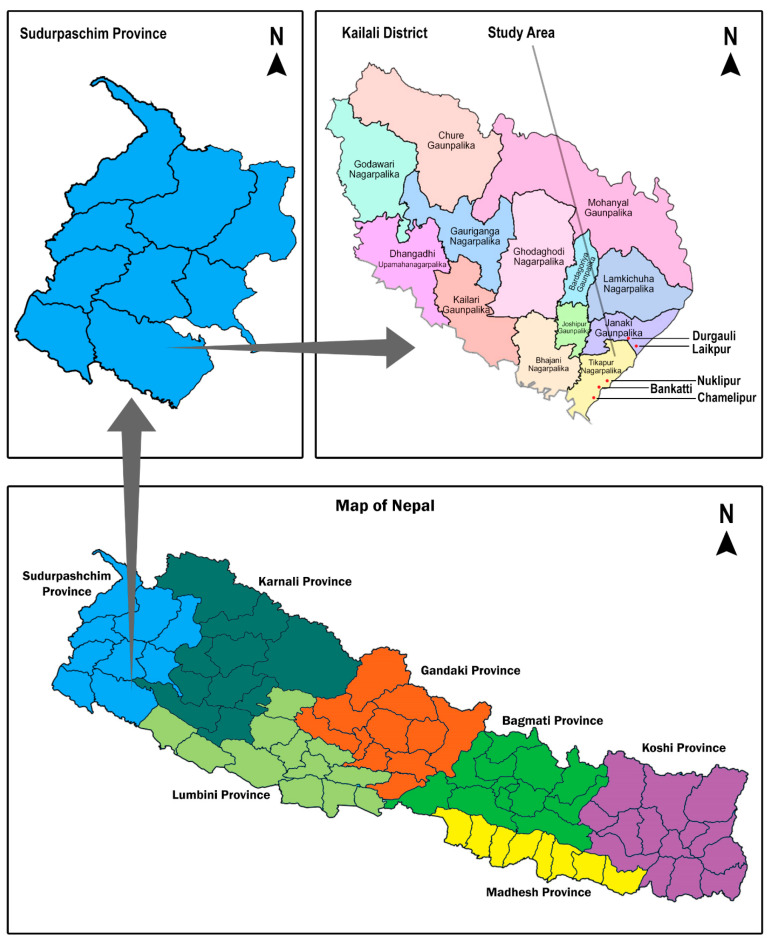
Map of Nepal indicating the study area.

**Table 1 healthcare-12-01099-t001:** Baseline characteristics of research participants.

Items (N = 141)		Frequency	Percentage (%)
Age (In Years) Mean ± SD	51.18 ± 10.15		
Caste	Nepali	38	27%
	Tharu	103	73%
Religion	Christian	12	8.5%
	Hindu	129	91.5%
Area	Nuklipur	47	33.3%
	Bankatti	6	4.3%
	Chamelipur	29	20.6%
	Durgauli	50	35.5%
	Laikpur	9	6.4%

SD: standard deviation; N: number; %: percentage.

**Table 2 healthcare-12-01099-t002:** Women participants’ responses regarding the purpose of participation in the functional literacy education program.

Items (N = 141)	Frequency	Percentage (%)
Purpose of participating in the literacy program		
For my personal adaptation and development	102	72.3
For a happy home life	10	7.1
For the development of the community and the village	27	19.1
To increase the quality of work life	2	1.4
Others	0	0

N: number; %: percentage.

**Table 3 healthcare-12-01099-t003:** Women participants’ opinions toward satisfaction with the functional literacy education program in rural settings of the Tikapur area, measured using the Likert scale.

S. No	Satisfaction Response with the Functional Literacy Education Program (N = 141)	Very Low N (%)	Low N (%)	Moderate N (%)	High N (%)	Very High N (%)
**A.** **Training course**						
1.	The objectives and contents of the training course are useful for my life	0 (0)	2 (1.40)	33 (23.4)	65 (46.1)	41 (29.1)
2.	The training course helps me improve income	0 (0)	0 (0)	42 (29.8)	71 (50.4)	28 (19.9)
3.	The topics of the training course are in line with the objectives and purposes of the training course	0 (0)	3 (2.1)	40 (28.4)	66 (46.8)	32 (22.7)
**B.** **Facilities**						
4.	The education place (venue) is convenient to access	0 (0)	2 (1.4)	16 (11.3)	71 (50.4)	52 (36.9)
5.	Quality of the learning facilities and stationeries for the classroom	0 (0)	0 (0)	27 (19.1)	63 (44.7)	51 (36.2)
6.	Satisfaction of accommodations/learning environments	0 (0)	2 (1.4)	24 (17)	77 (54.6)	38 (27)
**C.** **Trainers (Teachers)**						
7.	Trainers’ preparation for their lectures regarding content and format	0 (0)	0 (0)	18 (12.8)	92 (65.2)	31 (22)
8.	Qualification of experience, knowledge, teaching method of trainers	0 (0)	0 (0)	13 (9.2)	88 (62.4)	40 (28.4)
9.	Communication with trainers and trainees	0 (0)	1 (0.7)	21 (14.9)	73 (51.8)	46 (32.6)
10.	The lectures are interesting and easy to understand	0 (0)	0 (0)	22 (15.6)	87 (61.7)	32 (22.7)
11.	The lectures inspire me to develop critical thinking and creativity to apply to my life	0 (0)	0 (0)	34 (24.1)	95 (67.4)	12 (8.5)
12.	Group discussion, skill and experience sharing activities are integrated into the lectures	0 (0)	2 (1.4)	19 (13.5)	100 (70.9)	20 (14.2)
**D.** **General overview of training**						
13.	This training course is particularly helpful for my life	0 (0)	0 (0)	35 (24.8)	63 (44.7)	43 (30.5)
14.	My satisfaction with the whole training course	0 (0)	2 (1.4)	47 (33.3)	65 (46.1)	27 (19.1)
15.	We really need more similar training courses to improve our capacity	0 (0)	0 (0)	18 (12.8)	24 (17)	99 (70.2)

N: number; %: percentage.

**Table 4 healthcare-12-01099-t004:** Women’s participants’ responses to the functional literacy education program in empowering livelihood.

Items (N = 141)	Frequency	Percentage (%)
**Overall effects of the education program**		
Ineffective	3	2.1
Somewhat effective	2	1.4
Very effective	136	96.5
**Future participation in the literacy program**		
I will not	12	8.5
I will participate if circumstances permit	31	22
I will surely participate next time	98	69.5
**Recommend program to neighbors in future**		
Yes	125	88.7
No	16	11.3
**Life changes after program**		
No changes	12	8.5
Changes	129	91.5
**Begin income-related work after program**		
Yes	74	52.5
No	67	47.5
**Literacy program aids community development**		
Yes	140	99.3
No	1	0.7
**Able to read and write after program**		
Yes	138	97.9
No	3	2.1
**Satisfaction with reading and writing skills after program**		
Very much	59	41.8
Moderate	79	56
A little	3	2.1
Not at all	0	0

N: number; %: percentage.

**Table 5 healthcare-12-01099-t005:** Women’s participants’ responses regarding the contributions of the functional literacy education program to knowledge about menstrual hygiene.

**Items (N = 141)**	**Frequency**	**Percentage (%)**
**Do you know about menstrual hygiene?**		
Yes	136	96.5
No	5	3.5
**Do you practice chhaupadi?**		
Yes	20	14.2
No	121	85.8
**Hygiene neglect during menstruation can cause illness**		
Strongly agree	25	17.7
Agree	97	68.8
Disagree	19	13.5
Strongly disagree	0	0
**Clean water and soap are essential for menstrual hygiene**		
Strongly agree	37	26.2
Agree	103	73
Disagree	1	0.7
Strongly disagree	0	0
**Handwashing before genital cleaning prevents reproductive infection**		
Strongly agree	26	18.4
Agree	101	71.6
Disagree	8	5.7
Strongly disagree	6	4.3
**Clean sanitary pad/reusable cloth are essential for menstrual hygiene**		
Strongly agree	55	39
Agree	81	57.4
Disagree	5	3.5
Strongly disagree	0	0
**What do you use to clean reusable cloth?**		
With soap and water	111	78.7
Water only	30	21.3
**How do you dispose menstrual materials?**		
Open field	11	7.8
Latrine	24	17
Put it in the bin	106	75.2
**Has participating in the literacy program improved your knowledge of menstrual hygiene?**		
Strongly agree	41	29.1
Agree	92	65.2
Disagree	8	5.7
Strongly disagree	0	0

N: number; %: percentage.

**Table 6 healthcare-12-01099-t006:** Women’s participants’ responses regarding the contributions of the functional literacy education program to knowledge about maternal and child health.

Items (N = 141)	Frequency	Percentage (%)
**Do you know about the frequency of minimum antenatal care (ANC) visits?**		
Yes	115	81.6
No	26	18.4
**Do you know pregnancy risk symptoms?**		
Yes	119	84.4
No	22	15.6
**Do you know about taking supplement medicine (Iron, Zinc, and Folic acid tablets) during pregnancy?**		
Yes	130	92.2
No	11	7.8
**Do you know about eating balanced diets during and after pregnancy?**		
Yes	136	96.5
No	5	3.5
**Do you know about the suitable age for pregnancy?**		
Below 18 years	13	9.2
After 18 years	128	90.8
**Do you know about child immunization?**		
Yes	108	76.6
No	33	23.4
**Do you think the literacy program has helped you to improve your knowledge of maternal and child health?**		
Strongly agree	31	22
Agree	106	75.2
Disagree	4	2.8
Strongly disagree	0	0
**Do you think hand washing is important before eating and after using toilets?**		
Strongly agree	75	53.2
Agree	66	46.8
Disagree	0	0
Strongly disagree	0	0
**Do you think the literacy program helps you to improve your knowledge of health and hygiene?**		
Strongly agree	38	27
Agree	103	73
Disagree	0	0
Strongly disagree	0	0

N: number; %: percentage.

**Table 7 healthcare-12-01099-t007:** Correlations (Spearman’s rho) between the measures of satisfaction level with the functional adult literacy education program based on the training course, facilities, trainers, and general overview of training.

			TC	F	T	GOT
Spearman’s rho	**TC**	Correlation Coefficient	1.00	0.402 **	0.428 **	0.456 **
		Sig. (2-tailed)	.	0.000	0.000	0.000
		N	141	141	141	141
	**F**	Correlation Coefficient	0.402 **	1.000	0.327 **	0.302 **
		Sig. (2-tailed)	0.000	.	0.000	0.000
		N	141	141	141	141
	**T**	Correlation Coefficient	0.428 **	0.327 **	1.000	0.290 **
		Sig. (2-tailed)	0.000	0.000	.	0.000
		N	141	141	141	141
	**GOT**	Correlation Coefficient	0.456 **	0.302 **	0.290 **	1.000
		Sig. (2-tailed)	0.000	0.000	0.000	.
		N	141	141	141	141

** The correlation is significant at the 0.01 level (2-tailed). TC indicates training course, F indicates facilities; T indicates trainers, and GOT indicates general overview of training.

**Table 8 healthcare-12-01099-t008:** Association between sociodemographic variables and satisfaction with functional literacy program by participants.

Items	Very Low N (%)	Low N (%)	Moderate N (%)	High N (%)	Very High N (%)	Total N	χ^2^	*p* Value
**Age (Years)**							19.806	0.003
20–40	0 (0)	0 (0)	4 (17.4)	14 (60.9)	5 (21.7)	23		
41–60	0 (0)	0 (0)	31 (33.3)	40 (43)	22 (23.7)	93		
61–80	0 (0)	2 (8)	12 (48)	11 (44)	0 (0)	25		
**Caste**							6.682	0.083
Nepali	0 (0)	0 (0)	8 (21.1)	24 (63.2)	6 (15.8)	38		
Tharu	0 (0)	2 (1.9)	39 (37.9)	41 (39.8)	21 (20.4)	103		
**Religion**							5.347	0.148
Hindu	0 (0)	2 (1.6)	44 (34.1)	56 (43.4)	27 (20.9)	129		
Christian	0 (0)	0 (0)	3(25)	9 (75)	0 (0)	12		
**Area**							23.662	0.023
Nuklipur	0 (0)	0 (0)	19 (40.4)	19 (40.4)	9 (19.1)	47		
Bankatti	0 (0)	1 (16.7)	2 (33.3)	2 (33.3)	1 (16.7)	6		
Chamelipur	0 (0)	0 (0)	12 (41.4)	11(37.9)	6 (20.7)	29		
Durgauli	0 (0)	0 (0)	11 (22)	29 (58)	10 (20)	50		
Laikpur	0 (0)	1 (11.1)	3 (33.3)	4 (44.4)	1 (11.1)	9		

N: number; %: percentage; χ^2^: chi-square.

## Data Availability

The data presented in this study are available in the article (tables).

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
