# Peer review of "Enhancing Health and Empowerment: Assessing the Satisfaction of Underprivileged Rural Women Participating in a Functional Literacy Education Program in Kailali District, Nepal"

_healthcare, 2024, doi:10.3390/healthcare12111099_

Round 1
Reviewer 1 Report
Comments and Suggestions for Authors
This is a very interesting article because of the reality it addresses and the program it analyses. However, some several important issues and problems should be resolved for the article to have the necessary quality and relevance.
- One relevant aspect is that it appears that the course being analyzed was implemented between 2013 and 2015. When was the questionnaire administered? I think some more information could be provided about the program itself: trainers, daily hours, approach... Also, there is no explanation as to why the course was not continued after 2015, this information would be pertinent to understand the context and design of the study.
- I have one concern about the sample: 141 women participated out of 255 participants in the courses. Despite 141 women is a valuable number and I agree on not “forcing” participation, the fact that those participating are “those who wanted to” (and represent the 55% of the participating women) may condition the results. We can assume that they are the most favorable and satisfied with the program... the article does not sufficiently address the limitations of this sample. The conclusions state, in fact, that the results obtained show that "functional literacy education programs" have profound effects on people's lives .... but it should be questioned to what extent this study can be applied to the results of "these programs" (any functional literacy program?), or even to the specific program being analyzed.
- In the presentation of the results, the explanations basically reproduce the information in the tables.
- Another important issue is the focus of the paper. According to the abstract, I expected the results to focus on the impact of the program on women's lives. In the results section, point 4 refers to "Contributions of the Functional Literacy Education Program in Empowering Women's Livelihood". However, the detail of the questions (and the answers) does not allow for an analysis of what these effects are. It is asked in generic terms whether "Have changes in your life after participating in the literacy program " but it is not detailed what these changes are.
- This has to do with a fundamental issue, which is the way in which social impact, or the effects of the programme, are understood. Some specific issues within “satisfaction” are related to “usefulness in meeting their needs”, (wich is related to immediate outputs, but not clearly to actual benefits/changes in their lives). Others, such as “satisfaction with facilities” may be relevant but it is not a key issue in regards to social impact. I am confident that the programme does indeed bring about positive changes in the lives of women and suggest the authors deepen the evaluation of the social impact.
- One of these changes is related to the important issue of menstrual hygiene. It is justified and very important, although there is no space to go deeper and it is not clear whether this is a central aspect or not of the article. Likewise, results show that many women participants in the program do not practice chhaupadi , but we do not know if this is also (or not) an effect of the program.
- In the Discussion section, where data from the study is provided, it reverts to the data presented in the previous section. I consider that this section can be improved selecting and focusing on the contributions of the paper to existing knowledge.
I think the paper is relevant and improving the context and the focus of the contributions would greatly improve the manuscript.
Author Response
Dear Reviewer,
Authors extend their sincere gratitude to the reviewer for precisely reviewing our manuscript and providing us invaluable feedback. Your careful examination, insightful suggestions, and constructive feedback have greatly enriched the quality of our work. We truly appreciate the time and effort you dedicated to evaluating our manuscript. Based on your feedback, we have revised and resubmitted our manuscript. Please kindly find the attachment for point to point response. We are truly grateful to you.

Reviewer 2 Report
Comments and Suggestions for Authors
To the editors,
The study is relevant as it investigates an important topic - the impact of functional literacy education programs on underprivileged rural women in Tikapur, Nepal.
I’d recommend the authors to revise/restructure several places for improving the manuscript.
Research questions: While the authors outline the study's objectives as assessing the outcomes of the education program (Lines 109-130), it would be beneficial to include explicit research questions and hypotheses for the current study. For example, research questions could aim to verify correlations between measures of the training program or assess satisfaction based on geographic location.
Method. Please clarify the “convenience sampling technique” (Line 153) or cite a reference related to this technique. The manuscript mentions that 141 underprivileged women from five rural villages near Tikapur were selected for the study. However, it does not elaborate on the criteria for participant selection or the characteristics of the villages and literacy centers run by the NGO. Providing more context on the selection process and setting would help readers better understand the study's scope and generalizability of findings.
Figure 1. The text on the current map lacks clarity and is difficult to read. Please revise the text format to enhance the readability. If possible, provide an additional map illustrating the locations of the five villages included in the study in Tikapur.
Table 1. Please add mean (SD) to the first raw (next to the Age (in years) or at the bottom of the Table.
Table 1 shows the distribution of samples by caste, religion and area. However, it lacks analyses related to these features. I strongly recommend the authors to consider further statistical tests to find out differences and associations between these characteristics and the program outcome domains. E.g. How different outcomes between the Nepali and Tharu.
Table 2. Please consider remove or summarize the text between the lines 236-240. As the results is clearly displayed in the Table, no need to repeat the same information in the preceding text.
Table 3. Similar to the above, please consider shortening or removing the text between Line 262-326, as is repeats the content shown in Table 3.
Table 3. Please consider reconstructing the Table 3. Put all Satisfaction responses (very low, low, medium, high, very high) in the columns instead of rows. It could shorten the table reserving the same amount of information. OR, consider to convert Likert-scale data to numeric values. Please don’t omit the values, even it is 0(0%). E.g. You omitted the value “Very low” for the “A.1. The objectives and contents of the training course are useful for my life.”
Table 4, Table 5, and Table 6. Similar to the above, please remove the redundant text and consider restructuring the table for better visibility.
Line 444. “Table 1” should be Table 7. Please remove the sentence “The positive correlation…” (lines 437-440), because it is obvious.
Discussion: Please consider removing, or moving the paragraphs related to the background to the Introduction part:
Lines 462-477. Background of education
Lines 522-535. Background of literacy program.
Lines 552-559. Background of MCH in Nepal.
Please add a paragraph to discuss the possible limitations of the study. (cross-sectional design, lack of control group. Bias related to relationship between the trainers and the participants etc.)
Author Response

(The authors gave the same response as above.)

Reviewer 3 Report
Comments and Suggestions for Authors
This is a well written a well-designed study that illustrates empowerment through literacy in women in Nepal
Some minor changes should be done:
1. Include some statistical results in your abstract.
2. In table 1, explain what is in age 51.18 ± 10.15. It is mean and SD as explained in the text, but a table should be able to be read and understood in isolation without reading the text.
3. In section 3.7 focus more in the magnitude of the correlation between the variables, than in the probability.
4. Please check the format of reference number 5. I think is incorrect https://www.undp.org/publications/global-trends-challenges-and-opportunities-implementation-sdgs
5. Reference number 26 is incomplete.
6. Reference number38 it is written in capital letters . Is that correct?
7. What does it means the “*” in reference number 40.
Author Response

(The authors gave the same response as above.)

Round 2
Reviewer 1 Report
Comments and Suggestions for Authors
I appreciate the authors’ effort for improving the manuscript.
In regards to the administation of the questionnaire (and the whole evaluation of the programme): now it is clarified that it was administrated in 2024. This means 9 years after the programme implementation. This time span necessarily has to be considered as relevant when considering the study. It may not be a limitation if it is used to obtain relevant data on the changes that the program produced in the women. But in my opinion, the questionnaire does not seem to be designed to identify and evaluate a 9 year ago program and its effects. There are very specific questions on issues such as methodology or materials used that may not have been necessary now, after so many time (or the answers to which may be questionable). On the other hand, questions about the changes produced are left in rather broad statements (e.g. "life changes after programme" - yes/No"), missing the opportunity to go deeper into this. Likewise, the specific questions on the contributions of the functional education literacy program to knowledge about menstrual hygiene give us information on women's practices and knowledge in 2024, but not on the extent to which these practices have been influenced by the program. Authors are aware that study does not directly assess the causal relationship between program participation and the cessation of chhaupadi. But then, the findings about practices and knowledge about menstrual hygiene does not clearly fit with the purpose of the study. It is undoubtedly a very important issue, but to present these findings it in the context of this article seems a bit out of place. Moreover, sentences like “In our study, majority of women reported participating in functional literacy program have improved their knowledge on menstrual hygiene” are inaccurate.
I am afraid these limitations are not correctable with explanations in the paper, and that they affect the relevance and pertinence of the results presented.
Author Response
Authors extend their sincere gratitude to the reviewer for precisely reviewing our manuscript and providing us invaluable feedback. Your careful examination, insightful suggestions, and constructive feedback have greatly enriched the quality of our work. We truly appreciate the time and effort you dedicated to evaluating our manuscript. Based on your feedback, we have revised and resubmitted our manuscript.

Reviewer 2 Report
Comments and Suggestions for Authors
The authors responsively addressed all comments and revised the manuscript accordingly. I have no further comments.
Author Response
The authors would like to express their sincere gratitude to the reviewer for dedicating time to review the revised manuscript. Your contribution has been invaluable to this process. We are thankful for your acknowledgment that we
have satisfactorily addressed all your previous comments. Your feedback is helpful in ensuring the quality and clarity of our work. If you have no further comments, we consider this feedback as crucial for the refinement of our research. We're grateful for your thorough review and constructive input throughout this process.
